# QuaDUE-CCM: Interpretable Distributional Reinforcement Learning using Uncertain Contraction Metrics for Precise Quadrotor Trajectory Tracking

**Yanran Wang**
Imperial College London
yanran.wang20@imperial.ac.uk

**James O'Keeffe**
Imperial College London
j.okeeffe@imperial.ac.uk

**Qiuchen Qian**
Imperial College London
qiuchen.qian19@imperial.ac.uk

**David Boyle**
Imperial College London
david.boyle@imperial.ac.uk

**Abstract:** Accuracy and stability are common requirements for quadrotor trajectory tracking systems. Designing an accurate and stable tracking controller remains challenging, particularly in unknown and dynamic environments with complex aerodynamic disturbances. We propose a Quantile-approximation-based Distributional-reinforced Uncertainty Estimator (QuaDUE) to accurately identify the effects of aerodynamic disturbances, i.e., the uncertainties between the true and estimated Control Contraction Metrics (CCMs). Taking inspiration from contraction theory and integrating the QuaDUE for uncertainties, our novel CCM-based trajectory tracking framework tracks any feasible reference trajectory precisely whilst guaranteeing exponential convergence. More importantly, the convergence and training acceleration of the distributional RL are guaranteed and analyzed, respectively, from theoretical perspectives. We also demonstrate our system under unknown and diverse aerodynamic forces. Under large aerodynamic forces ($>2\ m/s^2$), compared with the classic data-driven approach, our QuaDUE-CCM achieves at least a 56.6% improvement in tracking error. Compared with QuaDRED-MPC, a distributional RL-based approach, QuaDUE-CCM achieves at least a 3 times improvement in contraction rate.

**Keywords:** Quadrotor trajectory tracking, Learning-based control

## 1 Introduction

Designing a precise and safe trajectory tracking controller for autonomous Unmanned Aerial Vehicles (UAVs), such as quadrotors, is a critical yet extremely different problem, particularly in unknown and dynamic environments with unpredictable aerodynamic forces. To achieve reliable trajectory tracking for agile quadrotor flights, two key challenges are: 1) solving for highly variable uncertainties caused by these aerodynamic forces; and 2) ensuring the tracking controller performs precisely and reliably under these dynamic uncertainties.

Previous research shows that aerodynamic effects deriving from drag forces and moment variations caused by the rotors and the fuselage [1] are the primary source of uncertainty, where these aerodynamic effects appear prominently at flight speeds greater than 5 $ms^{-1}$ in wind tunnel experiments [2]. The causes of these effects are extremely complex: combinations of individual propellers, airframe [3], rotor–rotor and airframe–rotor turbulent interactions [4], and other turbulent propagation [5]. These aerodynamic effects are therefore chaotic and difficult to model directly.

To address these concerns, two areas of research have emerged. The first uses a distributional Reinforcement Learning (RL) [6] method to interact with the complex and changeable uncertainties. The second extends the contraction theory [7] to a control-affine system, called Control Contraction Metric (CCM). This contraction certificate guarantees that the system will exponentially converge

6th Conference on Robot Learning (CoRL 2022), Auckland, New Zealand.

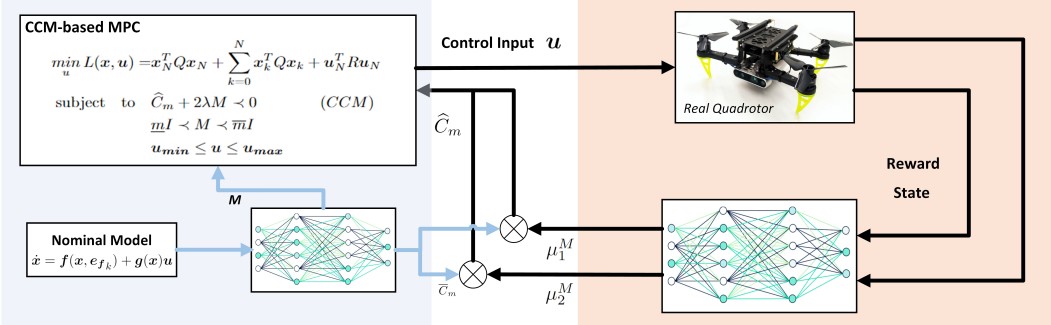

Figure 1: QuaDUE-CCM: a distributional-RL-based estimator and a CCM-based controller

to track any feasible reference trajectory [8]. In applications such as quadrotor tracking, we are concerned with the whole tracking process rather than just stabilizing to a fixed point. One common problem with certificate control approaches, like CCM-based control, is that they heavily rely on the accurate knowledge of the system model. Thus, when the model is uncertain, robust or adaptive versions must be considered.

We propose **Qua**ntile-approximation-based **D**istributional **U**ncertainty **E**stimatior for **C**ontrol **C**ontraction **M**etric (QuaDUE-CCM), as a precise, reliable and systematic quadrotor tracking framework for use with high variance aerodynamic effects. Our contributions can be summarized:

1) **QuaDUE**, a distributional-RL-based uncertainty estimator with quantile approximation that can sufficiently estimate variable uncertainties of contraction metrics. In the cases tested, we show that QuaDUE outperforms traditional RL, such as Deep Deterministic Policy Gradient (DDPG) [9], and prior Distributional RL approaches, such as C51 [6];

2) **QuaDUE-CCM**, a quadrotor trajectory tracking framework that integrates a distributional-RL-based uncertainty estimator into a CCM-based nonlinear optimal control problem;

3) **Theoretical understanding and mathematical proofs**, i.e., the convergence guarantee and acceleration analysis are provided for the interpretability of the distributional RL.

## 2   Related Work

**Quadrotor Uncertainty Modelling:**  Precise dynamics modeling in quadrotor autonomous navigation is challenging, since a great variety of aerodynamic effects, such as unknown drag coefficients and wind gusts, can be generated by agile flight in high speeds and accelerations. Data-driven approaches, such as Gaussian Processes (GP) [1, 10] and neural networks [11] combined with Model Predictive Control (MPC), have been shown accurate modelling of aerodynamic effects. Due to the nonparametric nature of GP, the GP-based approach hardly scale to the large datasets of complex environments [12]. While neural network-based approaches learn the nonlinear dynamical effects more accurately [13], achieving adaptability and robustness from these learning-based approaches are still ongoing challenges. One important reason is that these training datasets are collected from simulation and/or real-world historical records, from which it is difficult to fully describe the complexity in their environments [14].

**Reinforcement Learning for uncertainty:**  Compared with the existing GP-based and neural-network-based approaches, RL, an adaptive and interactive learning approach, is introduced to model highly dynamic uncertainties in recent work [15]. Distributional RL constructs the entire distributions of the action-value function instead of the traditional expectation, where, to some extent, it addresses the key challenge of traditional RL, i.e., biasing the actions with high variance values in policy optimization [9]. Since some of these values will be overestimated by random chance [16], such actions should be avoided in risk-sensitive or safety-critical applications such as autonomous quadrotor navigation. More importantly, recent work [17] shows that the distributional RL has better interpretability, where we can understand or even accelerate the entire RL training

process. This is an encouraging trend, especially for safety-critical applications such as quadrotor autonomous navigation, towards closing the gap between theory and practice in distributional RL.

**Control Certificate Techniques:** Control certificate techniques guarantee the synthesis of control policies and certificates [18], where the controller can optimize control policies whilst ensuring the satisfaction of the certificate properties. A Control Lyapunov Function (CLF)-based controller [19] ensures that the system state is Lyapunov stable, and a Control Barrier Function (CBF)-based controller [20] ensures that the system state is maintained in defined safety sets given by the barrier function. Contraction is a property of the closed-loop system [7]. [21] reformulates the CCM certificate as a Linear Matrix Inequality problem and solves using Sum-of-Squares (SoS). However, the SoS-based approach cannot extend to general robot systems since an assumption is that the system dynamics need to be polynomial equations or can be approximated as polynomial equations. Then [22, 23, 24] propose a contraction-metric-based control framework, which extends neural networks to certificate learning for contraction metrics. Moreover, based on the framework proposed by Tsukamoto et al., [8, 25, 26, 27, 28] are used to address higher dimensional control problems.

## 3 Preliminaries and notations

**Nominal Quadrotor Dynamic Model and its Control-affine Form:** the quadrotor is assumed as a six Degrees of Freedom (DoF) rigid body of mass $m$, i.e., three linear motions and three angular motions [1]. Different from [29, 30], the aerodynamic effect (disturbance) $e_f$ is integrated into the quadrotor dynamic model as follows [10]:

$$\dot{P}_{WB} = V_{WB} \qquad \dot{V}_{WB} = g_W + \frac{1}{m}(q_{WB} \odot c + e_f)$$
$$\dot{q}_{WB} = \frac{1}{2}\Lambda(\omega_B)q_{WB} \qquad \dot{\omega}_B = J^{-1}(\tau_B - \omega_B \times J\omega_B) \tag{1}$$

where $P_{WB}$, $V_{WB}$ and $q_{WB}$ are the position, linear velocity and orientation expressed in the world frame, and $\omega_B$ is the angular velocity expressed in the body frame (more detailed definitions please see [10]). Then we reformulate Equation 1 in its control-affine form: $\dot{x} = f(x, e_{fk}) + g(x)u + w$, Where $f : \mathbb{R}^n \mapsto \mathbb{R}^n$ and $g : \mathbb{R}^n \mapsto \mathbb{R}^{n \times m}$ are assumed by the standard as Lipschitz continuous [31, 8]. $x = [P_{WB}, V_{WB}, q_{WB}, \omega_B]^T \in \mathbb{X}$, $u = T_i \in \mathbb{U}$ and $w \in \mathbb{W}$ are the state, input and additive uncertainty of the dynamic model, where $T_i$ is the thrust of the $i$-th ($i \in [0, 3]$) motor, $\mathbb{X} \subseteq \mathbb{R}^n$, $\mathbb{U} \subseteq \mathbb{R}^{n_u}$ and $\mathbb{W} \subseteq \mathbb{R}^{n_w}$ are compact sets as the state, input and uncertainty space, respectively. $c$ is the collective thrust $c = [0, 0, \sum T_i]^T$. $e_{fk}$ is the aerodynamic effect estimated by VID-Fusion [32] in the $k$-th timestamp. Given an input : $\mathbb{R}^{\geq 0} \mapsto \mathbb{U}$ and an initial state $x_0 \in \mathbb{X}$, our goad is to design a quadrotor tracking controller $u$ such that the state trajectory $x$ can track *any* reference state trajectory $x_{ref}$ (satisfied the quadrotor dynamic limits) under a bounded uncertainty $w : \mathbb{R}^{\geq 0} \mapsto \mathbb{W}$.

**Control Contraction Certificates:** most existing methods on quadrotor trajectory tracking have so far been demonstrated in stabilization, i.e., a fixed-point-tracking controller [31, 33, 14]. These fixed-point-tracking controllers are relatively efficient in a low-speed and low-variance trajectory. However, to track a high-speed and high-variance trajectory in agile flight, we need a broader specification [34] rather than Lyapunov guarantees that simply stabilize in a fixed point.

Contraction theory [7] gives an analysis on the convergence evolution between the pairs of close trajectories in nonlinear systems, i.e., incremental stability. When extended to a control-affine system, the change rate is defined as $\delta\dot{x} = A(x, u)\delta x + B(x)\delta u$, where $A(x, u) := \frac{\partial f(x)}{\partial x} + \sum_{i=1}^m u^i \frac{\partial b_i}{\partial x}, B(x) = g(x)$, $b_i$ is the $i$-th vector of $B$, and $u^i$ is the $i$-th element of $u$. Just as a CLF is an extension from the idea of Lyapunov function $V(x) : \mathbb{X} \mapsto \mathbb{R}$, the contraction theory has been extended to a CCM $M(x) : \mathbb{X} \mapsto \mathbb{R}^{n \times n}$, i.e., a distance measurement between neighboring trajectories (a contraction metric) that shrinks exponentially, for tracking controller design [35, 36, 37]. As the matrix-valued function $M(x)$ denotes a symmetric and continuously differentiable metric, $\delta x^T M \delta x$ defines a Riemannian manifold [38, 7]. If there exists $M \succ 0$ (uniformly positive definite) and exponential convergence of $\delta x$ to 0, the Riemannian infinitesimal length will converge to 0, then the system is contraction. We have the following theorem:

**Theorem 1** *(Contraction Metric* [7, 34]*)*: The Lie derivative $\partial_v(\cdot)$ of a matrix-valued function $M(x_m)$ along a vector $v \in \mathbb{R}^n$ is $\partial_v M := \sum_{i=1}^n \frac{\partial M}{\partial x_m^i}\dot{v}^i$. In a control-affine system, if

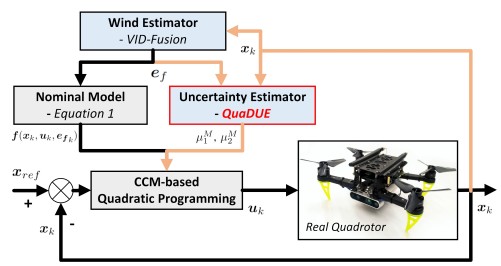
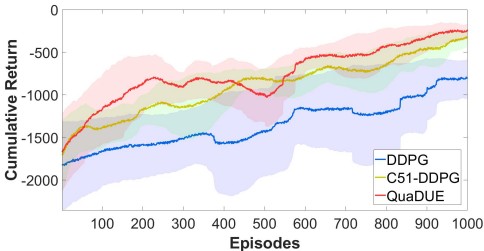

Figure 2: The position of QuaDUE-CCM in the Quadrotor system:

Figure 3: Learning curves of RL algorithms. The simulated speed is set as 0.4.

there exists: $\dot{M} + \text{sym}(M(A + BK)) + 2\lambda M \prec 0$, then the inequality $\|\boldsymbol{x}(t) - \boldsymbol{x}_{ref}(t)\| \leq Re^{-\lambda t} \|\boldsymbol{x}(0) - \boldsymbol{x}_{ref}(0)\|$ with $\forall t \geq 0$, $R \geq 1$ and $\lambda > 0$ holds, where $A = A(\boldsymbol{x}, \boldsymbol{u})$ and $B = B(\boldsymbol{x})$ are defined above, and $\dot{M} = \partial_{f(\boldsymbol{x})+g(\boldsymbol{x})u} M = \sum_{i=1}^{n} \frac{\partial M}{\partial x_{m}^{i}} \dot{x}^i$, $\text{sym}(M) = M + M^T$, and $\boldsymbol{x}_{ref}(t)$ is a sequential reference trajectory. Therefore, we say the system is contraction. (Proof see Supplementary part A)

**Overview of the Control Framework:** the overall framework is shown in Fig. 2. The aim of this work is to design a quadrotor controller, which combines wind estimation - i.e., VID-Fusion [32], for tracking the reference trajectory $\boldsymbol{x}_{ref}(t)$ of the nominal model (Equation 1) under highly variable aerodynamic disturbances. We propose a CCM-based MPC, where the parameter $\theta$ corresponds to weights of a neural network that estimates the uncertainty $\Delta_1$ and $\Delta_2$ in the CCM dynamics. Combining with the neural network, a Distributional-RL-based Estimator, i.e., QuaDUE, is used to learn the uncertainties adaptively whilst satisfying the constraints of the contraction metric.

## 4 Distributional Reinforcement Learning for CCM

**Quantile-approximation-based Distributional-reinforced Uncertainty Estimator:** in policy evaluation setting, given a 5-tuple Markov Decision Process [39]: $MDP := \langle S, A, P, R, \gamma \rangle$ (where $S$, $A$, $P$, $R$ and $\gamma \in [0, 1]$ are the state spaces, action spaces, transition probability, immediate reward function and discount rate), a distributional Bellman equation defines the state-action distribution $Z$ and the *Bellman operator* $\mathcal{T}^\pi$ as [6, 40]: $\mathcal{T}^\pi Z(\boldsymbol{s}, \boldsymbol{a}) \overset{D}{:=} R(\boldsymbol{s}, \boldsymbol{a}) + \gamma Z(\boldsymbol{s}', \boldsymbol{a}')$, where $\boldsymbol{s} \in S$ and $\boldsymbol{a} \in A$ are the state and action vector. $\pi$ is a deterministic policy under the policy evaluation. In policy control setting, based on the quantile approximation [40], the distributional *Bellman optimality operator* $\mathcal{T}$ is defined as: $\mathcal{T}Z(\boldsymbol{s}, \boldsymbol{a}) \overset{D}{:=} R(\boldsymbol{s}, \boldsymbol{a}) + \gamma Z(\boldsymbol{s}', \arg\max_{a'} \mathbb{E}_{\boldsymbol{p}, R} [Z(\boldsymbol{s}', \boldsymbol{a}')])$ where

$Z_\theta(\boldsymbol{s}, \boldsymbol{a}) := \frac{1}{N} \sum_{i=1}^{N} \delta_{q_i(\boldsymbol{s}, \boldsymbol{a})} \in Z_Q$ is a quantile distribution mapping one state-action pair $(s, a)$ to a uniform probability distribution supported on $q_i$. $Z_Q$ is the space of quantile distribution within $N$ supporting quantiles. $\delta_z$ denotes a Dirac with $z \in \mathbb{R}$. Then the maximal form of the Wasserstein metric is proved to be a contraction in [6] as:

$$\bar{d}_\infty(\Pi_{W_1} \mathcal{T}^\pi Z_1, \Pi_{W_1} \mathcal{T}^\pi Z_2) \leq \bar{d}_\infty(Z_1, Z_2) \tag{2}$$

where $W_p$, $p \in [1, \infty]$ denotes the $p$-Wasserstein distance. $\Pi_{W_1}$ is a quantile approximation under the minimal 1-Wasserstein distance $W_1$. The maximal form of the $p$-Wasserstein metric is defined as $\bar{d}_p := \sup W_p(Z_1, Z_2)$. Equation 2 illustrates that, for a fixed policy $\pi$, the Bellman operator $\mathcal{T}^\pi$ over value distribution $Z_1$ and $Z_2$ is a contraction in the maximal form of the Wasserstein metric, which is an important support for the convergence guarantee below, i.e., **Proposition 2** *(Policy Evaluation)*.

**Distributional-RL-based Estimator for CCM Uncertainty:** denoting $\dot{M} + \text{sym}(M(A + BK))$ in **Theorem 1** by $\overline{C}_m(\boldsymbol{x}, \boldsymbol{x}_{ref}, \boldsymbol{u}_{ref}, \theta_{nominal-metric})$, where $\theta_{nominal-metric}$ is the parameter of the neural network in the contraction metric learning [8] for the nominal function Equation 1 with

$e_f = 0$, then the truth $C_m$ under uncertainty becomes: $C_m = \sum_{i=1}^n \frac{\partial M}{\partial x^i} \dot{x}^i + \text{sym}(M(A + BK)) + \Delta_1^M(x) + \Delta_2^M(x)u = \overline{C}_m + \Delta_1^M(x) + \Delta_2^M(x)u$ .

We employ an agent of QuaDUE, the detailed algorithm for which is given in Supplementary Algorithm 1, to estimate the uncertain terms in $\widehat{C}_m$: $\Delta_1^M(x)$ and $\Delta_2^M(x)$. Therefore, an estimation $\widehat{C}_m$ is constructed by: $\widehat{C}_m = \overline{C}_m + \mu_{\theta_{rl},1}^M(x) + \mu_{\theta_{rl},2}^M(x)u$, where $[\mu_{\theta_{rl},1}^M(x), \mu_{\theta_{rl},2}^M(x)]$ - i.e., a symmetric matrix - is the output (action) of QuaDUE. $\theta_{rl}$ is again the neural network parameters. Then our goal of the Distributional-RL-based Estimator - i.e., QuaDUE - is clear: learn a policy $\mu_1^M$, $\mu_2^M$ such that the estimation $\widehat{C}_m$ as close as to the true value $C_m$. Notice here that we use the same neural network $\theta_{rl}$ for $\mu_{\theta_{rl},1}^M(x)$ and $\mu_{\theta_{rl},2}^M(x)$. In Section 5 below, we will demonstrate our reasoning and give proof.

A framework of QuaDUE is illustrated in Fig.1. The QuaDUE learns a policy which combines with the CCM uncertainties $\Delta_1^M(x)$, $\Delta_2^M(x)$ and other dynamic constraints. Then we design the reward function as $R_{t+1}(s, a, \theta) = R_{contraction}(s, a, \theta) + R_{track}(s, a)$, where $\theta \in \Theta$ represents the neural network parameters. Thus, the reward function is defined in detail as:

$$R_{contraction}(s, a, \theta) = -\omega_{c,1}[\underline{m}I - M]_{ND}(s) - \omega_{c,2}[M - \overline{m}I]_{ND}(s)$$
$$-\omega_{c,3}[\widehat{C}_m + 2\lambda M]_{ND}(s, a, \theta) \qquad (3)$$
$$R_{track}(s, a) = -(\boldsymbol{x}_t(s, a) - \boldsymbol{x}_{ref,t})^{\mathrm{T}} H_1(\boldsymbol{x}_t(s, a) - \boldsymbol{x}_{ref,t}) - \boldsymbol{u}_t^{\mathrm{T}}(s, a)H_2\boldsymbol{u}_t(s, a)$$

where $\underline{m}$, $\overline{m}$ are hyper-parameters, $H_1$ and $H_2$ are positive definite matrices, and $[A]_{ND}$ is for penalizing positive definiteness where $[A]_{ND} = 0$ iff. $A \prec 0$, and $[A]_{ND} \geq 0$ iff. $A \succeq 0$.

These QuaDUE outputs, i.e., $\mu_{\theta_{rl},1}^M(x)$ and $\mu_{\theta_{rl},2}^M(x)$, are then fed into the CCM constraints derived from the nominal model. Specifically, the estimated $\widehat{C}_m$ is used as the best guess of $C_m$ for MPC to satisfy the true constraints. The MPC formulation combining with CCM constraints are shown in Fig. 1, where the learned uncertainties are used to construct the contraction constraints of the MPC. Then the MPC is specified to a quadratic optimization formulation and implemented using CasADi [41] and ACADOS [42].

**Certified Learning on a Contraction Metric for the Nominal Model:** as shown in Fig. 1, a certified learning controller proposed in [34] is constructed to learn a Contraction Metric for the nominal model (Equation 1). An expert control policy is employed by solving a nonlinear MPC problem [10]. The contraction metric $M$ and control policy $\overline{\mu}$ is represented as two neural networks [8], where the output of the metric network A and the metric is achieved $M = sym(A)$. The parameters of the metric and policy networks are optimized simultaneously in the training process, where the loss function is: $L = L_{metric} + L_{policy} = (1/N_{train})(\sum_i [\underline{m}I - M]_{ND}(\boldsymbol{x_i}) + \sum_i [M + \overline{m}I]_{ND}(\boldsymbol{x_i}) + \sum_i [\widehat{C}_m + 2\lambda M]_{ND}(\boldsymbol{x_i}, \boldsymbol{\mu_i}) + (1/N_{train}) \sum_i \|\boldsymbol{\mu_i} - \boldsymbol{\mu_{expert,i}}\|$.

## 5 Properties of QuaDUE: Convergence and Acceleration analysis

In this section, the properties of the proposed Distributional-RL-based Estimator - i.e., QuaDUE - are analyzed, including convergence guarantees and training acceleration analysis. In particular, we empirically verify a more stable and accelerated convergence process than traditional RL algorithms (shown in Section 6), and demonstrate a theoretical understanding of the proposed QuaDUE.

**Convergence Analysis of QuaDUE:** Equation 2 shows that the *Bellman operator* $\mathcal{T}^\pi$ is a $p$-contraction under the $p$-Wasserstein metric $\overline{d}_p$ [6]. The Wasserstein distance $W_p$ between a distribution $Z$ and its Bellman update $\mathcal{T}^\pi Z$ can be minimized iteratively in Temporal Difference learning. Therefore, we present the convergence analysis of QuaDUE for *Policy Evaluation* and *Policy Improvement*, respectively.

**Proposition 2** *(Policy Evaluation [40, 14])*: Given a deterministic policy $\pi$, a quantile approximator $\Pi_{W_1}$ and $Z_{k+1}(\boldsymbol{s}, \boldsymbol{a}) = \Pi_{W_1} \mathcal{T}^\pi Z_k(\boldsymbol{s}, \boldsymbol{a})$, the sequence $Z_k(\boldsymbol{s}, \boldsymbol{a})$ converges to a unique fixed point $\widetilde{Z}_\pi$ under the maximal form of $\infty$-Wasserstein metric $\overline{d}_\infty$. See Supplementary part A for Proof.

**Proposition 3** *(Policy Improvement [14])*: Denoting an old policy by $\boldsymbol{\pi_{old}}$ and a new policy by $\boldsymbol{\pi_{new}}$, there exists $\mathbb{E}[Z(s, a)]^{\boldsymbol{\pi_{new}}}(s, a) \geq \mathbb{E}[Z(s, a)]^{\boldsymbol{\pi_{old}}}(s, a), \forall s \in \mathcal{S}$ and $\forall a \in \mathcal{A}$. See Supplementary part A for Proof.

Based on **Proposition 2** and **Proposition 3**, we are ready to present **Theorem 4** to demonstrate the convergence of QuaDUE.

**Theorem 4** *(Convergence)*: Denoting the policy of the $i$-th policy improvement by $\boldsymbol{\pi^i}$, there exists $\boldsymbol{\pi^i} \to \boldsymbol{\pi^*}$, $i \to \infty$, and $\mathbb{E}[Z_k(s,a)]^{\boldsymbol{\pi^*}}(s,a) \geq \mathbb{E}[Z_k(s,a)]^{\boldsymbol{\pi^i}}(s,a)$, $\forall s \in \mathcal{S}$ and $\forall a \in \mathcal{A}$. See Supplementary part A for Proof.

**Training Acceleration of QuaDUE:** training acceleration is investigated for our QuaDUE in order to guarantee a more stable and predictable RL training process. For implementation execution, we use Kullback-Leibler (KL) divergence [43] instead of Wasserstein Metric. KL divergence is proven and implemented 'usable' in [6]. Thus we use KL divergence to analyze the acceleration in RL training process, which is closer to the real implementation.

Similar to [14], we denote $J_\theta(s,a) = D_{KL}(p^{s,a}, q_\theta^{s,a})$ as the *histogram distributional loss*, where $p^{s,a}(x)$ and $q_\theta^{s,a}$ are the true and approximated density function of $Z(s,a)$. We denote $E_J(\theta) = \mathbb{E}_{(s,a)\sim\rho_\pi}[J_\theta(s,a)]$ is the expectation of $J_\theta$, where $\rho_\pi$ is the generated distribution under the deterministic policy $\pi$. Since the unbiased gradient estimation of KL divergence, there exits $\mathbb{E}_{(s,a)\sim\rho_\pi}[\|\nabla J_\theta(p_\mu^{s,a}, q_\theta^{s,a}) - \nabla E_J(\theta)\|^2] = \kappa\sigma^2$, where we represent the approximation error between $p_\mu^{s,a}$ and $q_\theta^{s,a}$. A good degree of the approximation leads to a small $\kappa$, which then effects the training acceleration of the distribution RL [17], as shown in **Theorem 5**.

Next we define a stationary-point for the derivative of $E_J(\theta)$ [17]: if there exits $\|\nabla E_J(\theta_T)\| \leq \tau$ ($\tau \in (0,1)$), the updated parameters $\theta_T$ after $T$ steps is a first-order $\tau$-stationary point. Denote F(s) as the feature on each state $s$ and let the support of $Z(s,a)$ have $k$ partitions. Assuming $\|F(s)\| < l$, we immediately present **Theorem 5** for the training acceleration.

**Theorem 5** *(Training Acceleration)*: In the training process of the distribution RL (i.e., QuaDUE):

1) Let sampling steps $T = 4kl^2 E_J(\theta_0)/\tau^2$, if there exists $\kappa \leq 0.25\tau^2/\sigma^2$, the $J_\theta$ optimized by stochastic gradient descend converges to a $\tau$-stationary point.

2) Let sampling steps $T = kl^2 E_J(\theta_0)/(\kappa\tau^2)$, if there exists $\kappa > 0.25\tau^2/\sigma^2$, the $J_\theta$ optimized by stochastic gradient descend will not converge to a $\tau$-stationary point. (Proof see Supplementary part A)

**Theorem 5** demonstrates two cases of training acceleration in our QuaDUE. In (1) of **Theorem 5**, the convergence of QuaDUE is accelerated where there is a low approximation error $\kappa$ between $p^{s,a}(x)$ and $q_\theta^{s,a}$. In (2) of **Theorem 5**, a relatively larger $\kappa > 0.25\tau^2/\sigma^2$ will give no guarantee for acceleration convergence. But if $(1/T)\sum_{t=0}^{T-1} \mathbb{E}[\|\nabla E_J(\theta_t)\|^2] \leq 4\kappa\tau^2$ is satisfied, the distributional RL still converges to the stationary optimization point. This understanding from theoretical analysis is consistent with our empirical experiment results in Section 6: when we add small external forces (disturbances) in the specific scenarios Fig. 4, the performance of the two distributional RL - i.e., 'QuaDUE-CCM' and 'QuaDRED-MPC' - is less than 'DDPG-CCM'. The cases in [40] also verify this observation from simple Atari games.

## 6    Evaluation of Performance

To evaluate the performance of the proposed QuaDUE-CCM, we compare against the convergence performance of DDPG [9], C51-DDPG [6] in training process, the tracking performance of GP-MPC [1, 10], DDPG-CCM and QuaDRED-MPC [14] under highly variable aerodynamic disturbances (uncertainties). The parameters of our QuaDUE-CCM are summarized in Table 1 according to the benchmark works (see [1, 14, 44]). The parameters $\omega_{c,1}$, $\omega_{c,2}$ and $\omega_{c,3}$ in Equation 3 are set as $1e^{-4}$, $1e^{-4}$ and $1e^{-4}$ [8]. The matrices $H_1$ and $H_2$ in Equation 3 are chosen as $H_1 = diag\{2.5e^{-2}, 2.5e^{-2}, 2.5e^{-2}, 1e^{-3}, 1e^{-3}, 1e^{-3}, 2.5e^{-3}, 2.5e^{-3}, 2.5e^{-3}, 2.5e^{-3}, 1e^{-5}, 1e^{-5}, 1e^{-5}\}$ and $H_2 = diag\{1.25e^{-4}, 1.25e^{-4}, 1.25e^{-4}, 1.25e^{-4}\}$, respectively [14].

**Comparative Performance of QuaDUE Training:** we have two main training process: firstly, we learn a Contraction Metric for the nominal model (shown in Fig. 1), where both metric and policy networks are 2 hidden layers of 64 neurons and $N_{train} = 1e^4$. Then we operate a training process by generating external forces in RotorS [45], where the programmable external forces are in the horizontal plane with range [-3,3] ($m/s^2$). The training process has 1000 iterations where the quadrotor state is recorded at 16 $Hz$. The convergence curves of the training are illustrated in Fig. 3,

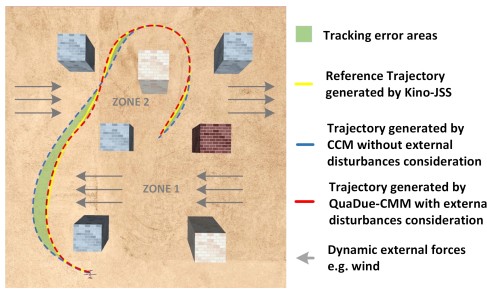
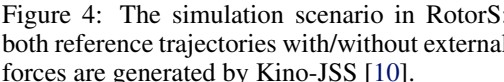
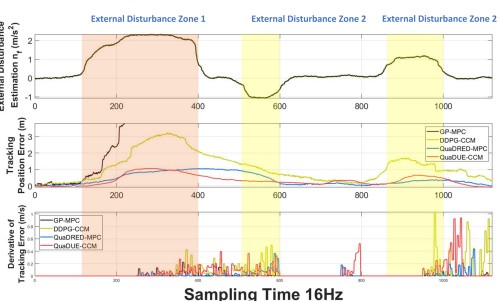

Figure 4: The simulation scenario in RotorS: both reference trajectories with/without external forces are generated by Kino-JSS [10].

Figure 5: Specific scenarios results: Wind estimation $\boldsymbol{n}_f$ (expressed in body frame), position error $(m)$ and derivatives of tracking error.

where we show the comparative training performance of DDPG, C51-DDPG and our QuaDUE. Our performance illustrates that the two distributional RL approaches, C51-DDPG and QuaDUE, outperform the traditional DDPG RL approach, whilst our proposed QuaDUE achieves the largest cumulative return. More importantly, our QuaDUE maintains the highest convergence speed, which again verifies the theoretical guarantees demonstrated in Section 5.

**Comparative Performance of QuaDUE-CCM in Variable Aerodynamic Disturbance:** we evaluate our proposed QuaDUE-CCM with variable aerodynamic forces generated in ZONE 1 and ZONE 2 of the specific scenario, as described in Fig. 4. We compare the tracking position errors and derivatives of tracking error with different and opposite heading aerodynamic forces with [0.0, 1.5, 0.0] $m/s^2$ and [0.0, -1.0, 0.0] $m/s^2$ in ZONE 1 and ZONE 2, respectively. As shown in Fig. 5, the three benchmarks are: 1) GP-MPC, a state-of-the-art trajectory tracking method [1, 10]; 2) DDPG-CCM, an adaptive approach extending from RL-CBF-CLF-QP [31] to solve the CCM uncertainties along with a nonlinear MPC; 3) QuaDRED-MPC, a novel learning-based MPC for uncertainties combining with a distributional RL agent and a nonlinear MPC [14].

In Fig. 5, the two distributional-RL-estimator-based frameworks (i.e., QuaDRED-MPC and QuaDUE-CCM) react to the sudden aerodynamic effects more sufficiently than the traditional RL-based estimator (i.e., DDPG). Among the two distributional-based methods, the QuaDRED-MPC's tracking performance is slightly better

Table 1: Parameters of QuaDUE-CCM

| Parameters | Definition | Values |
|---|---|---|
| $l_{\theta_a}$ | Learning rate of actor | 0.0015 |
| $l_{\theta_c}$ | Learning rate of critic | 0.0015 |
| $\theta_a$ | Actor neural network: fully connected with two hidden layers (128 neurons per hidden layer) | - |
| $\theta_c$ | Critic neural network: fully connected with two hidden layers (128 neurons per hidden layer) | - |
| $D$ | Replay memory capacity | $10^4$ |
| $B$ | Batch size | 256 |
| $\gamma$ | Discount rate | 0.9995 |
| - | Training episodes | 1000 |
| $T_s$ | MPC Sampling period | 50ms |
| $N$ | Time steps | 20 |

(accumulated error 6.73% less) than our proposed QuaDUE-CCM. This is mainly because the QuaDRED-MPC assesses the environment uncertainties and then directly feeds the uncertainty estimation into the nominal model while the QuaDUE-CCM is an indirect method required to satisfy the contraction metrics, which makes the tracking actions more conservative. However, based on the contraction theory, our proposed QuaDUE-CCM has 20.7% bigger derivative of tracking error than QuaDRED-MPC and the contraction rate is $Ce^{-\lambda t}$ with $C = 3.59$ and $\lambda = 1.046$. This again means that, under variable uncertainties, our QuaDUE-CCM converges to a reference trajectory in a higher contraction rate (exponential contraction) both with theoretical (contraction theory) and empirical guarantees.

Table 2: Comparison of Trajectory Tracking under Programmatic External forces

| Ex. Forces ($m/s^2$) | Method | Succ. Rate | Time (s) | Err. (m) | Contra. Rate ($\lambda/C$) |
|---|---|---|---|---|---|
| Z1: [0, 0.5, 0.0] Z2: [0, -0.5, 0.0] | GP-MPC | 100% | 23.54 | 3.44 | 0.097~0.63/6.49 |
| | DDPG + CCM | 97.6% | 24.11 | 3.82 | 0.285~1.01/3.54 |
| | QuaDRED-MPC | 95.3% | 25.54 | 4.30 | 0.100~0.64/6.41 |
| | QuaDUE-CCM | **95.1%** | **25.84** | **4.78** | **0.302~1.07/3.54** |
| Z1: [0, 2.5, 0.0] Z2: [0, -2.5, 0.0] | GP-MPC | 69.8% | 38.62 | 24.27 | 0.093~0.61/6.53 |
| | DDPG + CCM | 79.8% | 33.10 | 18.62 | 0.240~0.92/3.83 |
| | QuaDRED-MPC | 89.1% | 29.57 | 14.93 | 0.089~0.58/6.49 |
| | QuaDUE-CCM | **87.2%** | **30.19** | **15.49** | **0.278~1.01/3.63** |
| Z1: [-3.0, 3.5, 0.0] Z2: [-2.0, -2.0, 0.0] | GP-MPC | 0% | - | - | - |
| | DDPG + CCM | 48.5% | 43.64 | 27.24 | 0.257~0.81/3.85 |
| | QuaDRED-MPC | 83.3% | 36.40 | 17.66 | 0.079~0.53/6.69 |
| | QuaDUE-CCM | **84.1%** | **36.45** | **17.13** | **0.265~0.95/3.59** |

Then larger and more complex external forces, i.e., [-2.5, 2.5, 0.0] and [-3.0, 3.0, 0.0] ($m/s^2$), are generated in ZONE 1 and ZONE 2, respectively, as shown in Fig. 4. In Table 2, we compare the success rate, operation time, accumulated tracking error and contraction rate with the corresponding variable external forces in ZONE 1 and ZONE 2. Our results show that RL-based methods are not always better, especially with relatively small external forces. For example, the success rate of RL-based methods (i.e., 'DDPG + CCM', 'QuaDRED-MPC' and 'QuaDUE-CCM') is lower than 'GP-MPC'. We also find that the two distributional-RL-based methods have close tracking errors, where in some cases 'QuaDRED-MPC' is sightly better than 'QuaDUE-CCM'. However, compared to 'QuaDRED-MPC', our 'QuaDUE-CCM' achieves 302%, 312% and 335% improvements in contraction rate.

## 7    Conclusion

In this paper, we propose a precise and reliable trajectory tracking framework, QuaDUE-CCM, for quadrotors in dynamic and unknown environments with highly variable aerodynamic effects. QuaDUE-CCM combines a distributional-RL-based uncertainty estimator and CCMs to address large and variable uncertainties on quadrotor tracking. A Quantile-approximation-based Distributional-reinforced Uncertainty Estimator, QuaDUE, is proposed to learn the uncertainties of contraction metrics adaptively, where the convergence and acceleration in the RL training process are analyzed from theoretical perspectives. Based on the contraction theory, CCMs are used to guarantee exponential convergence to any feasible reference trajectory, which significantly improve the reliability of our tracking systems. We empirically demonstrate that our proposed approach can track agile trajectories with at least 36.2% improvement under large aerodynamic effects, where experimental data can favourably verify our theoretical results.

## 8    Limitation

As mentioned in Sections 5 & 6, in comparison with GP-MPC, our QuaDUE-CCM performs weakly under small disturbances. The potential reasons include QuaDUE-CCM's indirectly variable uncertainties modelling and the relatively conservative outputs caused by its theoretical guarantees. Our ongoing work concerns incorporating direct uncertainty estimation, i.e., adding uncertainties into the controller directly, and certificate-based uncertainty estimation to improve the performance under relatively small uncertainties. Another limitation is the computational complexity, where we only use 16 Hz for the CCM uncertainty estimation, whereas GP-MPC uses 50 Hz. To improve this, QuaDUE-CCM may be deployed on dedicated hardware, for example, via FPGA implementation. Ongoing work concerns evaluating performance in real-world flight tests, during which our system will face a greater variety of aerodynamic disturbances.

**Acknowledgments**

This work was partially supported by UKRI [grant number NE/T011467/1].

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
