# OpenReview forum: "QuaDUE-CCM: Interpretable Distributional Reinforcement Learning using Uncertain Contraction Metrics for Precise Quadrotor Trajectory Tracking"
_robot-learning.org/CoRL/2022/Conference — CoRL 2022 Poster_

### Official Review · Reviewer_VCFb · 2022-07-30

**Originality:** Good
**Technical Quality:** Good
**Clarity Of Presentation:** Good
**Impact:** 3

**Recommendation:**

Weak Accept: I recommend accepting the paper, but will not argue for my recommendation if the majority of other reviewers have a different opinion.

**Summary:**

This paper proposes: (1) QuaDUE, a distributional RL based uncertainty estimator that evaluates aerodynamic effects that are difficult to model directly; (2) QuaDUE-CCM, a quadrotor trajectory tracking system that incorporates QuaDUE into a MPC controller based on control contraction metric. Thorough theoretical analysis (in both the paper and the supplementary document) shows that the QuaDUE is guaranteed to converge, and training acceleration could be achieved when the approximation error of the learned distribution is low. In the experiment, it could be seen that within the shown learning period QuaDUE outperforms DDPG in terms of the learned cumulative return. In addition, compared to DDPG-CCM, QuaDUE-CCM exhibits higher success rates in trajectory tracking tasks when external forces in the simulated environment are large.

**Issues:**

How the nominal model is learned with the nominal function via contraction metric learning is not described in the paper, and I think it would be more clear to the audience if the methodology of contraction metric learning is included. Besides, in the experimental study of QuaDUE-CCM, some other method being compared include a nonlinear MPC controller, but no description about its formulation is given.

**Quality Of The Limitations Section:**

Limitations are addressed clearly

**Reviewer Expertise:**

3: The reviewer is fairly confident that the evaluation is correct

**Robotics Focus:**

Highly relevant to robotics but no hardware experiments

**Strengths And Weaknesses:**

Strengths: The theoretical analysis of convergence and training acceleration of QuaDUE demonstrates the method's capability with solid proof. The description of QuaDUE-CCM is comprehensive, and the supplementary document provides useful explanations about the quadrotor dynamic model, algorithm of kino-JSS and more details about the QuaDUE-CCM implementation.

Weaknesses: It's not clear that the learning curves shown in Figure 3 converge, hence I think it's not quite convincing to view the result as an evidence of the convergence guarantee described in theoretical analysis. Besides, QuaDUE shows similar learning performance as C51-DDPG, and QuaDUE-CCM has similar trajectory tracking performance as QuaDRED-MPC. Thus, experiment results seem to illustrate that the advantage of distributional RL methods in uncertainty estimation, and the proposed algorithms show a common performance among them.

**Summary Of Recommendation:**

This paper proposes a novel distributional RL based uncertainty estimator, and extends it to a quadrotor trajectory tracking system that is based on a control contraction metric based controller. Detailed theoretical analysis with respect to the convergence and training acceleration property of the proposed estimator is given. Simulation experiment results show that the proposed tracking system has higher success rates in trajectory tracking tasks with large external force than GP-MPC, a state-of-the-art trajectory tracking method, and DDPG-CCM, a system built with traditional RL.

Post-Rebuttal Comments: I appreciate the clarification provided by the authors, and I remain in favor of acceptance. Taking the authors' responses and the CoRL review criteria into account, I have decided to keep my rating of weak accept.

---

> ### Author Response · Authors · 2022-08-23
> **Response to Reviewer VCFb**
>
> Thank you for the detailed review and encouraging comments. We address your suggestions below. Please let us know if further clarification is needed.
>
> ***
>
> >* It's not clear that the learning curves shown in Figure 3 converge, hence I think it's not quite convincing to view the result as an evidence of the convergence guarantee described in theoretical analysis
>
> ***
>
> Thanks for your comment. The convergence and its convergence rate are guaranteed by **Theorem 4** and **Theorem 5** (Section 5). In Section 6, experimental results are used to validate the above theoretical analysis. The learning curves in Figure 3 show the trend of accumulative rewards within 1000 iterations, and is not intended as the evidence of convergence guarantees. We set iterations to 1000 based on the benchmark QuaDRED [1,2].
>
> ***
>
> >* QuaDUE shows similar learning performance as C51-DDPG, and QuaDUE-CCM has similar trajectory tracking performance as QuaDRED-MPC. Thus, experiment results seem to illustrate that the advantage of distributional RL methods in uncertainty estimation, and the proposed algorithms show a common performance among them.
>
> ***
>
> This is a good question! Yes, indeed, our proposed QuaDUE-CCM has similar tracking performance with its benchmark. However, among the two distributional-based methods (i.e., QuaDUE-CCM and QuaDRED-MPC), the QuaDRED-MPC's tracking performance is slightly better (accumulated error $6.73\%$ less) than our proposed QuaDUE-CCM.
>
>
>
> In Section 6, we will clarify as follows: 'This is mainly because the QuaDRED-MPC assesses the environments’ uncertainties and then directly feeds the uncertainty estimation into the nominal model, while the QuaDUE-CCM is an indirect method required to satisfy the contraction metrics, which makes the tracking actions more conservative. However, based on the contraction theory, our proposed QuaDUE-CCM has $20.7\%$ bigger derivative of tracking error than QuaDRED-MPC and the contraction rate is $C e^{-\lambda t}$ with $C=3.59$ and $\lambda=1.046$. This again means that, comparing with other distributional-based methods like QuaDRED-MPC, our QuaDUE-CCM converges to a reference trajectory with a higher contraction rate (exponential contraction) under variable uncertainties, both with theoretical (contraction theory) and empirical guarantees.'
>
> ***
>
> >* How the nominal model is learned with the nominal function via contraction metric learning is not described in the paper, and I think it would be more clear to the audience if the methodology of contraction metric learning is included.
>
> ***
>
> Thank you for highlighting this. This aligns with comments from Rev. 3. We will add a subsection 'Certified Learning on a Contraction Metric for the Nominal Model' in Section 4 to describe the construction and training process of the learned contraction $M$ and policy $\mu$ of the nominal model.
>
> ***
>
> >* in the experimental study of QuaDUE-CCM, some other method being compared include a nonlinear MPC controller, but no description about its formulation is given.
>
> ***
>
> We will add the reference for the benchmark we employed and compared: 1) GP-MPC, a combination between Guassian Process and a nonlinear MPC [3,4]; 2) DDPG-CCM, an adaptive approach extending from RL-CBF-CLF-QP to solve the CCM uncertainties along with a nonlinear MPC [5]; 3) QuaDRED-MPC, a novel learning-based MPC for uncertainties combining with a distributional RL agent and a nonlinear MPC [1].
>
>
>
> Reference:
>
>
>
> [1] Wang, Yanran, et al. "Interpretable Stochastic Model Predictive Control using Distributional Reinforced Estimation for Quadrotor Tracking Systems." arXiv preprint arXiv:2205.07150 (2022).
>
>
>
> [2] Zhang, Qingrui, Wei Pan, and Vasso Reppa. "Model-reference reinforcement learning for collision-free tracking control of autonomous surface vehicles." IEEE Transactions on Intelligent Transportation Systems (2021).
>
>
>
> [3] Wang, Yanran, et al. "KinoJGM: A framework for efficient and accurate quadrotor trajectory generation and tracking in dynamic environments." 2022 International Conference on Robotics and Automation (ICRA). IEEE, 2022.
>
>
>
> [4] Torrente, Guillem, et al. "Data-driven MPC for quadrotors." IEEE Robotics and Automation Letters 6.2 (2021): 3769-3776.
>
>
>
> [5] Choi, Jason, et al. "Reinforcement Learning for Safety-Critical Control under Model Uncertainty, using Control Lyapunov Functions and Control Barrier Functions." Robotics: Science and Systems (RSS). 2020.

---

### Official Review · Reviewer_PQAG · 2022-07-31

**Originality:** Good
**Technical Quality:** Very Good
**Clarity Of Presentation:** Good
**Impact:** 3

**Recommendation:**

Weak Accept: I recommend accepting the paper, but will not argue for my recommendation if the majority of other reviewers have a different opinion.

**Summary:**

This paper proposes a method for estimating uncertainties in quadrotor dynamics using distributional reinforcement learning and control contraction metrics (CCM). In particular, distributional RL is used to estimate the disturbances that the system is subjected to, and this is used in a CCM-based trajectory tracking control framework. The approach is demonstrated in simulation, and is shown to improve tracking error relative to baseline algorithms.

**Issues:**

See the comments above.

**Quality Of The Limitations Section:**

Limitations are addressed clearly

**Reviewer Expertise:**

4: The reviewer is confident but not absolutely certain that the evaluation is correct

**Robotics Focus:**

Highly relevant to robotics but no hardware experiments

**Strengths And Weaknesses:**

Strengths: The paper is generally well-written, mathematically solid, and provides an interesting synthesis between distributional RL and control contraction metrics which may be of interest for future research.

Comments, concerns, suggestions, and questions:
- There are some ambiguities in notation which sometime make it difficult to follow the math:
    - It seems $\lambda$ is overloaded in the proof of Theorem 5 (contraction rate vs. step size)?
    - It is unclear how $\Delta_1$, $\Delta_2$ factor into (1) or into the control affine form $\dot x = f(x) + B(x)u + w$.
    - $e_{fk}$ and $T_i$ are not defined
    - Lie derivative $\partial_{(\cdot)}$ is not defined
    - Several variables in Theorem 5 are not defined
- There is missing previous related work [1, 2] which also estimates worst-case disturbances in the context of learning-based motion planning to determine how they affect the tracking performance of a CCM-based controller.
- Related to the last note, it would be good to compare the proposed distributional RL approach to a “robust CCM” baseline [1, 2, 3], which attempts to do feedback control assuming worst-case disturbance, and the improvements in conservativeness afforded by using a distributional approach.
- How are $\omega_{c,i}$ chosen in the reward function? I couldn’t find information on that in the main body.
- In the QP in Figure 1, in general, it is difficult to guarantee a priori whether there exists a control input $u$ within the control limits $u_min$ and $u_max$ that satisfies the contraction condition, especially for a $\hat C$ which is being learned. In practice, do you run into feasibility problems in the QP? More comments clarifying this in the text would be helpful.
- In parts of the paper, to aid it would be good to have more intuition describing the math: it is unlikely that people are intimately familiar with both CCMs and distributional RL. For instance, before (2), you might describe what the “maximal form” is, and why it’s important that it has contractive properties, e.g., “the approximate Wasserstein distance between $Z_1$ and $Z_2$ shrinks upon applying the Bellman operator, which is important for convergence guarantees later on”.
- It would be good to have more details on the training of M, u, and the selection of the contraction rate \lambda.
- It would help to move some of the QuaDUE-CCM algorithm details from the appendix into the main body - right now the algorithm is described rather vaguely. More generally, in the main body of the paper, please reference the proofs and additional relevant material in appendix.

[1] Chou, Ozay, and Berenson. Model Error Propagation via Learned Contraction Metrics for Safe Feedback Motion Planning of Unknown Systems. CDC 2021.

[2] Chou, Ozay, and Berenson. Safe Output Feedback Motion Planning from Images via Learned Perception Modules and Contraction Theory. WAFR 2022.

[3] Singh, Landry, Majumdar, Slotine, and Pavone. Robust Feedback Motion Planning via Contraction Theory.

**Summary Of Recommendation:**

The paper is generally well-written and is an interesting combination of CCMs and distributional RL, which hopefully may inform a future line of research -- thus, I recommend acceptance.

---

> ### Author Response · Authors · 2022-08-23
> **Response to Reviewer PQAG**
>
> We appreciate the thoughtful, detailed and constructive feedback. Your concerns are addressed below. Please let us know if further clarification is needed.
>
> ***
>
> >* There are some ambiguities in notation which sometime make it difficult to follow the math:
>
> ***
>
> > It seems $\lambda$ is overloaded in the proof of Theorem 5 (contraction rate vs. step size)?
>
> We will modify the step size to $\iota$ instead of $\lambda$ to avoid ambiguity.
>
>
>
> >It is unclear how $\Delta_{1},\Delta_{2}$ factor into (1) or into the control affine form $\dot{{x}} = {f}({x}) + {B}({x}){u}+{w}$.
>
>
>
> $\Delta_{1}^{M},\Delta_{2}^{M}$ are the uncertain terms in $C_m$, where an implicit connection exists with additive uncertainty $w$ in the control affine form, i.e., additive uncertainty of the dynamic model $w$ leads to CCM uncertainty $\Delta_1^{M},\Delta_2^{M}$.
>
>
>
> >${e}\_{f k}$ and $T_i$ are not defined. Lie derivative $\partial_{v}(\cdot)$ is not defined. Several variables in Theorem 5 are not defined
>
>
>
> We will add the definitions of ${e}\_{f k}$ and $T_i$, which are the aerodynamic effect estimated by VID-Fusion in the $k$-th timestamp and the thrust of the $i$-th ($i\in[0,3]$) motor (in the body frame), respectively. $c$ is the collective thrust $c = [0,0,\sum T_i]^{\rm{T}}$.
>
> The Lie derivative $\partial_{v}(\cdot)$ of a matrix-valued function $M(x_m)$ along a vector $v\in\mathbb{R}^{n}$ is $\partial_{v}M:=\sum_{i=1}^n {\frac{\partial M}{\partial {x_m}^i}\dot{v}^i}$. We will also add the definition of $k,l,\tau,\sigma$ above Theorem 5.
>
> ***
>
> >* There is missing previous related work [1, 2] ... a CCM-based controller. Related to the last note, it would be good ... a distributional approach.
>
> ***
>
> Thanks for your suggestion. Papers [1,2] will be addressed in Section 2 of the final/CR manuscript.
>
> ***
>
> >* How are $\omega\_{c,i}$ chosen in the reward function? I couldn’t find information on that in the main body.
>
> ***
>
> The definition of $\omega\_{c,i}$ will be included in Section 6:  The parameters $\omega_{c,1}$, $\omega_{c,2}$ and $\omega_{c,3}$ in Equation 3 are set as $1e^{-4}$, $1e^{-4}$ and $1e^{-4}$. Previously they were in the Appendix.
>
> ***
>
> >* In the QP in Figure 1, in general, it is difficult to guarantee a priori ... More comments clarifying this in the text would be helpful.
>
> ***
>
> This is a very practical and constructive comment. Considering and meeting feasibility is part of our ongoing work. For hardware testing, we have 3.75:1 thrust to weight ratio on our quadrotor (the value is 3.5:1 in the simulated platform). This means that we have large enough range $u\_{min}$ and $u\_{max}$ to satisfy the contraction condition even in agile flight. However, the feasibility problem always exists and may be hard to avoid. There exists $\widehat{u}$ satifying the contraction condition whilst having minmum distance with $u\_{min}$/$u\_{max}$. In practice, ACADOS is used to solve the QP in the computation frequency $20$ Hz, where the input $u$ will be chosen as close as $\widehat{u}$ to converge exponentially. We will add clarifying comments accordingly, likely in the Limitations section.
>
> ***
>
> >* In parts of the paper, to aid it would be good to have more intuition describing the math:... convergence guarantees later on”.
>
> ***
>
> Thanks for the constructive suggestion! The 'maximal form' definition will be emphasized and we will also add some intuitive descriptions, e.g., 'Equation  2 illustrates that, for a fixed policy $\pi$, the Bellman operator $\mathcal{T}^{\pi}$ over value distribution $Z_1$ and $Z_2$ is a contraction in the maximal form of the Wasserstein metric, which is an important support for the convergence guarantee below, i.e., **Proposition 2** (Policy Evaluation).'
>
> ***
>
> >* It would be good to have more details on the training of M, u, and the selection of the contraction rate \lambda.
>
> ***
>
> This aligns with comments from Rev. 4. We will add a 'Certified Learning on a Contraction Metric for the Nominal Model' subsection to Section 4 to describe the construction and training process of the learned contraction $M$ and policy $\mu$ of the nominal model.
>
> ***
>
> >* It would help to move some of the QuaDUE-CCM algorithm details from the appendix into the main body - ... in appendix.
>
> ***
>
> This aligns with comments from Rev. 2, and we will revise to move proofs to the appendix. We will use the resulting available space to include more details about the algorithm and its implementation.

---

### Official Review · Reviewer_U1k2 · 2022-08-01

**Originality:** Good
**Technical Quality:** Good
**Clarity Of Presentation:** Fair
**Impact:** 2

**Recommendation:**

Weak Accept: I recommend accepting the paper, but will not argue for my recommendation if the majority of other reviewers have a different opinion.

**Summary:**

The authors propose the QuaDUE-CCM framework for robust trajectory tracking for quadrotors with highly variable uncertainties due to aerodynamic forces. The authors provide convergence guarantees as well as simulation results that demonstrate the effectiveness of the framework.

**Issues:**

See above

**Quality Of The Limitations Section:**

Limitations are addressed clearly

**Reviewer Expertise:**

1: The reviewer's evaluation is an educated guess

**Robotics Focus:**

Highly relevant to robotics but no hardware experiments

**Strengths And Weaknesses:**

The paper provides good experimental results that compare QuaDUE-CCM against state-of-the-art methods as well as convergence guarantees thoroughly supported by proofs. Unfortunately, I was unable to follow most of the proofs. Personally, I think the paper would have been easier to follow if it focused more on formulation of the framework and guarantees, and moved the proofs to the appendix. A section referring to how QuaDUE-CCM applies to other systems would help the paper as well.

**Summary Of Recommendation:**

The experimental results and guarantees seem thorough. However, I found the paper very confusing, mainly due to my lack of expertise, but also due to the dense and hard to follow writing.

---

> ### Author Response · Authors · 2022-08-23
> **Response to Reviewer U1k2**
>
> Thank you for the positive feedback. Your concerns are addressed below. Please let us know if further clarification is needed.
>
> ***
>
> >* Personally, I think the paper would have been easier to follow if it focused more on formulation of the framework and guarantees, and moved the proofs to the appendix.
>
> ***
>
> Thanks for your comment. This aligns with Rev. 3. Proofs will be moved and cited in the Appendix. We will also add some details and cite the detailed algorithm of QuaDUE in the main body and Appendix.
>
> ***
>
> >* A section referring to how QuaDUE-CCM applies to other systems would help the paper as well.
>
> ***
>
> Thanks for the suggestion! We will add a brief subsection in the final/CR manuscript to describe how QuaDUE-CCM performs in different systems. For example, according to the references [1-4], QuaDUE-CCM and similar certified-contraction-metric controllers can be applied on a planar vertical-takeoff-vertical-landing (PVTOL) [1], quadrotor [1,3], ground-based vehicle like Segway robot [1,2]. As our proposed QuaDUE-CCM employs a distributional-RL-based uncertainty estimator, the computational complexity needs to be considered when implemented in mobile autonomous systems. This limitation is also added in Section 8 (Limitation), where QuaDUE-CCM will be deployed on dedicated chips like FPGA to improve the computational complexity.
>
>
>
> Reference:
>
>
>
> [1] Sun, Dawei, Susmit Jha, and Chuchu Fan. "Learning Certified Control Using Contraction Metric." Conference on Robot Learning. PMLR, 2021.
>
>
>
> [2] Dawson, Charles, Sicun Gao, and Chuchu Fan. "Safe Control with Learned Certificates: A Survey of Neural Lyapunov, Barrier, and Contraction methods." arXiv preprint arXiv:2202.11762 (2022).
>
>
>
> [3] Wang, Yanran, et al. "Interpretable Stochastic Model Predictive Control using Distributional Reinforced Estimation for Quadrotor Tracking Systems." arXiv preprint arXiv:2205.07150 (2022).
>
>
>
> [4] Zhang, Qingrui, Wei Pan, and Vasso Reppa. "Model-reference reinforcement learning for collision-free tracking control of autonomous surface vehicles." IEEE Transactions on Intelligent Transportation Systems (2021).

---

### Official Review · Reviewer_6Poj · 2022-08-01

**Originality:** Very Good
**Technical Quality:** Very Good
**Clarity Of Presentation:** Very Good
**Impact:** 4

**Recommendation:**

Strong Accept: I recommend accepting the paper and will argue for my recommendation even if other reviewers hold a different opinion.

**Summary:**

This paper considers the problem of fly drones with wind. Neural networks are used to learn the effects of aerodynamic disturbances, which are the uncertainties between the true and estimated Control Contraction Metrics (CCMs). An RL agent is used to learn a policy, which computes a correction to the nominal contraction metric. The corrected contraction metric is then used in an MPC framework to construct the contraints of the MPC. It The approach is evaluated in simulation scenarios.

**Issues:**

1. The purpose of \mu^M_1 and \mu^M_2 is to make \hat{C}_m as close as to the true value C_m. If I understand correctly, the third item in R_{contraction} is for this purpose. But I don't understand how it works. This item can only make sure that \hat{C}_m \prec - 2\lambda M, but it does not imply that \hat{C}_m is close to C_m.
2. In line 137, it seems there should be a Z_{\theta}.

**Quality Of The Limitations Section:**

Limitations are addressed clearly

**Reviewer Expertise:**

3: The reviewer is fairly confident that the evaluation is correct

**Robotics Focus:**

Highly relevant to robotics but no hardware experiments

**Strengths And Weaknesses:**

Strengths:
1. Theoretical results.
2. Experiments in a simulation scenario.

Weakness:
1. No hardware experiments.
2. Concerns about the design of the reward function. (See Issues)

**Summary Of Recommendation:**

Although no hardware experiments were conducted, the authors presented strong theoretical results and experimental results in simulation.

---

> ### Author Response · Authors · 2022-08-23
> **Response to Reviewer 6Poj**
>
> Thank you for the thoughtful and constructive feedback. Your concerns are addressed below. Please let us know if further clarification is needed.
>
> ***
>
> >* The purpose of $\mu^M_1$ and $\mu^M_2$ is to make $\hat{C}m$ as close as to the true value $C_m$. If I understand correctly, the third item in R{contraction} is for this purpose. But I don't understand how it works. This item can only make sure that $\hat{C}_m \prec - 2\lambda M$, but it does not imply that $\hat{C}_m$ is close to $C_m$.
>
> ***
>
> Thanks for your question - there are three variables: $\overline{C}_m$, ${C}_m$ and $\widehat{C}_m$.
>
>
>
> $\overline{C}_m=\dot{M}+{\rm{sym}}(M(A+BK))$ represents the contraction metric for the nominal function (Equation 1 and its control form); $C\_m=\overline{C}\_m+\Delta\_{1}^{M}(x)+\Delta\_{2}^{M}(x)u$ is the true value of the contraction metric (nominal function + uncertainty); and $\widehat{C}_m$ is an estimation of the truth, i.e., $C\_m$. In practice, what we can achieve is $\overline{C}_m$ and $\widehat{C}_m$, while $C_m$, can only be obtained (and estimated) by $\widehat{C}_m$.
>
>
>
> Yes - the third item in $R\_{contraction}$ only makes sure that $\hat{C}_m \prec - 2\lambda M$. For the '$\hat{C}_m$ close to $C_m$', this is addressed by the interaction (or the feedback) within the environment. Whether '$\hat{C}_m$ is close to $C_m$ is judged by the whole trajectory tracking performance, i.e., $R\_{track}$. Thus, the $R\_{contraction}$ works for whether the estimated value $\hat{C}_m$ is satisfying the contraction metric, i.e., $\hat{C}_m \prec - 2\lambda M$ whilst $R\_{track}$ is for 'measuring the distance' between $\hat{C}_m$ and $C_m$ by interaction.
>
>
>
> Interaction is the essence of RL. In our QuaDUE-CCM, the dynamic uncertainties are modelled to give an optimal estimation $\hat{C}_m$ for $C_m$.
>
> ***
>
> >In line 137, it seems there should be a Z_{\theta}.
>
> ***
>
> Thanks for the comment! We will disambiguate the writing: a quantile distribution $Z_{\theta}\in Z_Q$ maps each state-action pair $(s,a)$ to a uniform probability distribution supported on ${\theta\_i}(s,a)$. $Z$ represents the whole distribution. Thus, in the Bellman operator function, it would be better to write $Z$ here. [1] can also be referred to for more information. Thanks!
>
>
>
> Reference:
>
>
>
> [1] Dabney, Will, et al. "Distributional reinforcement learning with quantile regression." Proceedings of the AAAI Conference on Artificial Intelligence. Vol. 32. No. 1. 2018.

---

### Meta-Review · Area_Chair_WL5c · 2022-09-04

**Recommendation:** Accept (Poster)
**Confidence:** 4

**Metareview:**

The reviewers evaluated this paper positively as it contains a solid theory (proof of convergence) for a new algorithm that combines distributional RL and control contraction metrics . The experiments are applied to a quadrotor trajectory following problem and showed good performance. One reviewer advocated an oral presentation, but I would rather recommend a poster presentation here due to missing hardware experiments and a rather narrow evaluation of the algorithm (only one domain).

**Best Paper Nomination:**

No